# Comparative Analysis of the Quality in Ripe Fruits of Cuiguan Pear from Different Regions

**DOI:** 10.3390/molecules28041733

**Published:** 2023-02-11

**Authors:** Miaoqiang Zhang, Bing Bai, Lei Chen, Haiyan Liu, Qiqi Jin, Liang Wang, Tao Feng

**Affiliations:** 1College of Life Science and Technology, Xinjiang University, Urumqi 830046, China; 2Institute of Quality Standard and Testing Technology for Agro-Products, Shanghai Academy of Agricultural Sciences, Shanghai 201403, China; 3School of Perfume and Aroma Technology, Shanghai Institute of Technology, Shanghai 201418, China

**Keywords:** Cuiguan pear, physicochemical parameters, volatile organic compounds, PCA, gas chromatography-ion mobility spectrometry (GC-IMS)

## Abstract

The Cuiguan pear is called “June snow” and the skin is thin; the meat is crisp and juicy; the taste is thick and fresh; and the juice is rich and sweet. In this study, the volatile organic compounds and the sensory and physicochemical parameters of the Cuiguan pear from four different regions of China (Sichuan (SC), Shangdong (SD), Chongming (CM), Zhuanghang (ZH)) were assessed. The highest differences in the physicochemical parameters were observed between four regions. The volatile fingerprints of GC-IMS showed great differences in the volatile of the Cuiguan pear, which suggested that the aroma of pears could be largely impacted by origin areas. (E)-ethyl-2-hexenoate can be used to distinguish between the ‘CM’ and pears from other regions. High contents of 2-heptanone, 1-pentanol, 1-butanol, 3-methylbutanol, butyl 2-methylbutanoate, heptyl acetate and butyl acetate were observed in the ‘SD’. Dimethyl trisulfide, 6-methyl-5-hepten-2-one, 3-hydroxy-2-butanone, 1-penten-3-one, beta-pinene, γ-terpinene, propanal, (e)-2-pentenal, (e)-2-heptenal, 1-pentanol and 3-methyl-1-pentanol were primarily contained in the ‘ZH’. Principal component analysis showed that there was very good discrimination based on the information obtained from GC-IMS for four samples. These findings were in agreement with the sensory analysis. In the opinion of the respondents to the consumer test, ‘ZH’ resulted in the most appreciated sample based on the average scores of the acceptability. This study provides some reference for the development and utilization of the Cuiguan pear.

## 1. Introduction

The pear is one of the most important temperate fruit tree species with high economic value [1]. Based on evolutionary, morphological and geographical characteristics, pears are divided into European pears and Asian pears, which contain a great range of nutrient elements and possess a crisp flesh, a high sugar content, and a stronger aroma and flavor [2]. China is one of the origin centers of pear species and pears are the third most important fruit in China [3]. The ‘Cuiguan pear’ (*Pyrus pyrifolia* Nakai.) is a famous pear cultivar in China and is widely cultivated in Southern China [4].

Fruit aroma is generated by a large number of non-volatile plant precursors through biochemical pathways, and is a key factor in evaluating fruit quality and affects acceptance by humans [5,6]. In a recent study, ethyl and hexyl acetates are the principal volatile organic compounds in pears [7]. A large body of evidence has demonstrated that fruit aroma is affected by the region, variety, maturity and other factors and the quality of pears in different regions is different [8]. However, there have been few studies on the volatile compounds and physicochemical parameters of Cuiguan pears with different origins.

In previous studies, GC-MS and an electronic nose were used to determine flavor or volatile components [9,10]. In recent years, gas chromatography–ion mobility spectrometry (GC–IMS) has been applied in food flavor analysis due to its advantages in rapid analysis, high sensitivity and variable volume injection with no pretreatment [11]. In this study, GC-IMS coupled with principal component analysis (PCA), physicochemical parameters, and sensory was employed to evaluate and compare four different regions of pears. This work could offer recommendations for future breeding efforts in the production of Cuiguan pears with improved nutritional and aroma quality.

## 2. Results and Dissussion

### 2.1. Chemical and Physical Parameters in Cuiguan Pear

In Table 1 results of the physicochemical parameters are reported. The skin color of a pear is considered an important factor in quality and consumer acceptance [12]. The results showed that the skin colors of the four Cuiguan pears were strong statistical differences (*p* < 0.05). The ‘SD’ had the highest lightness in four samples. The hue angle and chroma showed differences among the pears. The ‘SD’ and ‘ZH’ are greener than the other pears according to the statistical analysis of hue angle. Chroma ‘C’ expression is analogous to color saturation or intensity [13]. The ‘SC’ and ‘ZH’ had the highest color saturation. Firmness, TSS and TA are important factors for the eating quality of a pear and correlates with consumer acceptance [14,15,16]. The ‘CM’ and ‘ZH’ had better performances than the other pears with TSS and TA. Vitamin C content differed considerably in the four pears. The ‘CM’ and ‘ZH’ had higher Vitamin C contents (6.88 mg/100 g and 5.80 mg/100 g). The total phenol (TP) contents of the four pears ranged from 1.20 mg/g to 2.83 mg/g of FW, with the highest value in the ‘SC’ and the lowest value in the ‘SD’. In the same way, the total flavonoid (TFA) content of the four pears ranged from 0.62 mg/g to 1.26 mg/g in Table 1, with the highest content in the ‘ZH’ and the lowest content in the ‘SD’. The TP and TFA of a pear may increase the antioxidative activity for humans in blood plasma [17].

### 2.2. Mineral Element in Cuiguan Pear

Trace mineral elements are important not only for human health but also for plants [18]. In Table 2, the minerals of the pears are sorted by content, which are K, Mg, Ca, Na, B, Fe, Mn, Cu and Zn. The content of mineral elements in Cuiguan pears from different regions was significantly different (*p* < 0.05). In the ‘ZH’ it was observed that there were larger amounts of B, Mg, K, Cr, Mn, Fe, Ni, Zn and Mo. The ‘CM’ contained significantly higher levels of the mineral elements such as B, Na, Mn, Co and Cu. The ‘SD’ contained the highest mineral content of Se. In the same way, the ‘SC’ contained larger amounts of mineral elements such as Ca, Ti, V and Sn. Overall, the macro element mineral content of the pears were K, Mg, Ca and Na, and for the micro element mineral content were Fe, B, Zn, Mn, Cu, Cr, Ni, Mo, V, Se, Co and Ti. The results were in accord with [19].

### 2.3. HS-GC-IMS Topographic Plots of Cuiguan Pear

The retention time (RT), the ion migration time (DT) and peak intensity of gas chromatography were used in the qualitative analysis of the volatile components of the Cuiguan pears from different regions. Then, three-dimensional spectra (Figure 1A) and two-dimensional spectra (Figure 1B) were obtained with the Reporter plug-in. In Figure 1A,B, the background of the GC-IMS spectra was blue and the red vertical line at abscissa 1.0 was the reactive ion peak (RIP) after normalization. Each point on both sides of the RIP peak represented a volatile compound. The color expressed the concentration of the compound, with white indicating a lower concentration and red indicating a higher concentration.

In Figure 1B, most of volatile peak signals of the four Cuiguan pears were observed in ranges of retention time 100 to 800 s and drift time 1.0 to 2.0 ms. In Figure 1C, the differential plots were obtained by topographic plot deduction, with the topographical plot of the ‘SC’ taken as the background to obtain the difference comparison topographic plots. The blue point reflected a lower concentration of compounds than the reference and the red color meant a higher concentration of a volatile compound than that in the reference. In Figure 1C, there were a lot of blue (labeled with bule rectangles in Figure 1C) and red points (labeled with red rectangles in Figure 1C) that were observed in the ‘SD’, ‘CM’ and ‘ZH’, which showed a significant difference in the concentration of volatile compounds among the four Cuiguan pears from different regions.

### 2.4. Fingerprint Analysis of Volatile Compounds in Cuiguan Pear

In Figure 2, the fingerprints of VOCs in the Cuiguan pears were obtained by a Gallery Plot plug-in. Each row represented the whole peak signals of one sample and each column showed the same peak signals in different samples. The same as before, the color reflected the content of volatile compounds; the brighter the color is, the higher the content [20]. A total of 69 volatile compounds were identified by GC-IMS in the whole fingerprint spectrum. Among these volatiles, 50 compounds were identified by the GC-IMS Library and the NIST database because some compounds have the monomer (M) and dimer (D), including 25 esters, 8 alcohols, 8 aldehydes, 5 ketones, 2 terpenes, 1 thiophene and 1 sulfide (Table 3). In particular, some compounds could produce two peak signals because of the high proton affinity and concentration of the volatile compounds [21].

As shown in Figure 2, five volatile compounds where concentration was not a significant difference in all the Cuiguan pears were identified, including ethanol, ethyl-acetate (D and M), hexyl-acetate (D and M), butyl-acetate (M) and thiophene. A total of 28 compounds were an advantage in the ‘ZH’, including 15 esters, 4 alcohols, 4 ketones, 3 aldehydes, beta-pinene and dimethyl trisulfide. In particular, dimethyl trisulfide, 6-methyl-5-hepten-2-one, 3-hydroxy-2-butanone, 1-penten-3-one, beta-pinene, γ-terpinene, propanal, (e)-2-pentenal, (e)-2-heptenal, 1-pentanol and 3-methyl-1-pentanol were primarily contained in the ‘ZH’, which revealed low contents of other samples (labeled with yellow rectangles in Figure 2). (E)-ethyl-2-hexenoate were primarily contained in the ‘CM’ (labeled with green rectangles in Figure 2). As for the ‘SD’, 2-heptanone, 1-pentanol, 1-butanol, 3-methylbutanol, butyl 2-methylbutanoate, heptyl acetate and butyl acetate were identified as major volatile compounds. Five compounds were primarily contained in the ‘SC’, including (e)-2-Pentenal, heptanal, hexanal, isobutyl acetate and thiophene. The volatile compounds in the Cuiguan pears from different regions were basically similar. However, their signal intensities were significantly different.

Esters are synthesized by the oxidation of the fatty acid produced and contribute mainly to the fruits’ sweet aroma [22,23] and were observed in all the detected Cuiguan pear samples. Esters are considered to be the first aroma contributor due to their low-odor threshold [24]. Hexyl acetate, ethyl hexanoate, ethyl butanoate and ethyl acetate were the most plentiful esters in all the pears. Laia et al. [25] found that esters and alcohols are the main volatile compounds in the ‘Conference’ pear. This conclusion is consistent with our findings. Alcohols and aldehydes are derived from the metabolism of fatty acids and contribute mainly to the grassy and sweet [20,26]. Aldehydes had a higher content in the ‘ZH’ and ‘SC’. The aromas produced by the different concentrations of aldehydes vary, with high concentrations giving fatty and nutty aromas and low concentrations producing pleasant green odors [27]. For example, hexanal has an apple, leaf and subtle aroma; heptanal has a nutty and fruity green aroma; and nonanal has a strong fatty and sour flavor [21]. Alcohols are mainly produced by the oxidation of linoleic acid degradation products while saturated alcohols have a high odor threshold and contribute little to the overall aroma. Five types of ketones were identified in the Cuiguan pears from the four different areas, which usually have a long-lasting floral scent [21]. Terpene compounds are important components of a fruit’s aroma, giving the fruit a floral aroma [28]. In this study, two terpenes were identified; beta-pinene had a resin-like aroma and thiophene was similar to a citrus and lemon aroma. Chen et al. [7] found that 335 volatile organic compounds, mainly esters, alcohols, aldehydes and ketones, were identified from the ripe fruits of 12 western pear cultivars. This conclusion is generally consistent with our findings. Based on the volatile profiles characterized by GC-IMS, it is possible to distinguish the Cuiguan pears from different habitats.

### 2.5. PCA Analysis

PCA (Principal Component Analysis) analysis is a commonly used statistical analysis to reveal possible differences between samples by reducing the dimensions of large data sets [29]. In order to evaluate the difference among the aroma profiles of the Cuiguan pears from different regions, the peak location and peak intensity of volatile substances in the Cuiguan pear samples were obtained by GC-IMS and the difference in volatile substances content in the Cuiguan pear samples was analyzed using Dynamic PCA. The total contribution of the two principal components reached 88% (PC1 accounting for 67% and PC2 accounting for 21% of cumulative variance contribution). The cumulative contribution rate is more than 60%, showing that PCA as the separation model has a good effect [20]. In Figure 3, Cuiguan pears from different areas are separated from each other and Cuiguan pear samples from the same area are close to each other. This indicated that the volatile components of Cuiguan pears from different geographical areas were a significant difference, and GC-IMS combined with PCA analysis could achieve a better separation effect for Cuiguan pears from different regions.

### 2.6. Sensory Evaluation for Cuiguan Pear

In Figure 4, the results showed that the sensory description of Cuiguan pears from different areas was significantly different (*p* < 0.05). The consumers expressed a positive acceptance for the ‘ZH’ and ‘CM’. It is important to note that there was no statistically significant difference in the acceptance of the ‘ZH’ and ‘CM’. The ‘ZH’ was characterized by the best sweetness, fresh and graininess intensity, followed by the ‘CM’, which exhibited a slight lower value. The main attributes that influence consumer preferences are taste and aroma; texture; shape; and visual appearance [30]. In each of the previous indicators, the ‘ZH’ had a good performance, which may be the reason for the highest consumer acceptance.

## 3. Materials and Methods

### 3.1. Plant Materials

The Cuiguan pears were sampled from commercial pear orchards located in, Cangxi, Sichuan province, China; Laiyang, Shandong province, China; Chongming, Shanghai city, China; and Zhuanghang, Fengxian, Shanghai city, China in July 2022. All pears were collected at maturity stage and samples were named as SC (Sichuan Cuiguan pear), SD (Shandong Cuiguan pear), CM (Chongming Shanghai Cuiguan pear) and ZH (Fengxian Shanghai Cuiguan pear). The fresh collected pear samples were transported on ice and 50.0 kg of each sample was stored at 4 °C until instrumental or sensory analysis.

### 3.2. Physical and Chemical Parameters

The skin color of the pears were defined using a Minolta colorimeter (CR-400; Minolta, Konicaminolta, Japan) to evaluate the chromaticity values of L* (Lightness), a* (green to red) and b* (blue to yellow) on the equatorial sections of pear per cultivar. The hue angle (H*) and chroma (C) were calculated as reported by [13]. Firmness (expressed as newton) was measured as the force needed to reduce fruit diameter by 8 mm using a fruit firmness tester (MODEL GY-4 China).

Chemical parameters were assessed according to [18,31] with slight modification. The total soluble solid (TSS) content was measured with a digital refractometer and expressed as °Brix. The total acid (TA) content was measured using sodium hydroxide (Sodium hydroxide titrant, Shanghai Institute of Measurement and Testing Technology) titration. TA content was expressed as mg malic acid/g of fresh weight. Vitamin C was determined using redox titration with iodine solution and expressed as mg of vitamin C per 100 g of pear. ICP-MS (Agilent, America) was used for mineral composition analysis. Measurements were carried out on three technical replicates.

Total phenol (TP) and total flavonoid content (TFA) were measured with the Folin–Ciocalteu method and the aluminium chloride colorimetric method according to [8] with slight modification. The flesh for each pear (5 g) was cut and homogenized in 30 mL of ethanol (70%, v/v) and assisted by ultrasonic extraction. The mixture was centrifuged at 4 °C for 10 min. The supernatant was filtered and used for the subsequent analyses. TP was expressed as mg of catechol equivalents per 1 g of fresh weight (FW). TFA was expressed as mg of rutin equivalents per 1 g of fresh weight (FW). All measurements were carried out on three technical replicates.

### 3.3. Volatile Organic Compounds Analysis

Volatile organic compounds (VOCs) of the Cuiguan pears were analyzed by HS-GC-IMS, according to [28] with a little modification. It was composed of an Agilent 490 gas chromatograph (Agilent Technologies, Palo Alto, CA, USA) and an IMS instrument (FlavourSpec^®^, Gesellschaft für Analytische Sensorsysteme mbH, Dortmund, Germany) that was equipped with a PAL3 automatic sampler (CTC Analytics AG Company, Basel, Switzerland).

Flesh samples were obtained 0.5 cm under the skin of the pear at the upper, middle and lower parts of the pear and then tested after being crushed evenly. The pear flesh samples (2.0 g) were transferred into a 20-mL headspace vial and incubated at oscillating heating mode (40 °C) with the speed of 500 rpm for 20 min. Then the headspace was injected using a PAL3 sampler automatically with an injection volume of 500 μL and injector temperature of 65 °C. Then, VOCs were separated using a wax capillary column (30 m × 0.53 mm, 1.0 µm film thickness, RESTEK, Commonwealth of Pennsylvania, America) with column temperature fixed at 60 °C in GC. High purity nitrogen (≥99.999%) was used as a carrier gas. The initial flow rate was 2.0 mL/min for 2 min and increased to 10 mL/min within 8 min, and then increased to 100 mL/min within 10 min and maintained at 100 mL/min for 5 min. High purity nitrogen (≥99.999%) was used as a drift gas with a flow rate of 150 mL/min. VOCs were ionized in the IMS ionization chamber and ions were driven to the 9.8 cm drift tube with the nitrogen flow at a temperature of 45 °C. All measurements were carried out on three replicates. The retention index (RI) of each compound was calculated using n-ketones C4-C9 (Sinopharm Chemical Reagent Beijing Co., Ltd., Beijing, China) as external references. The identification of volatile compounds by comparing the RI and drift time with the GC-IMS library, and the content of volatile compounds was quantified based on the peak intensity in HS-GC-IMS. The volatile fingerprints of the pear samples were generated by The Reporter plug-in and the Gallery Plot plug-in.

### 3.4. Sensory Analysis

Each pear was cut into an average of four portions for consumer testing (avoid Browning, the maximum retention is 30 min after cutting), and each sample was randomly handed to the evaluator.

Before the test and after eating a sample, the evaluators were given salt-free crackers and mineral water to restore taste. The evaluators, all from the Shanghai Academy of Agricultural Sciences, had no prior training in sensory assessment and were made up of 70 people aged 22 to 60. Seven attributes of the Cuiguan pears, including sweetness, acidity, juiciness, graininess, chewiness, firmness, freshness and the acceptance of each pear was evaluated according to five-point scale (0–1, weaker; 1–2, weak; 2–3, middle; 3–4, strong; 4–5, stronger). The above method is slightly modified from a previous study [30,32]

### 3.5. Statistical Data Analyses

The results were statistically evaluated by SPSS Version 20.0 with the one-way analysis of variance (ANOVA) and the Student–Newman–Keuls (SNK) test for mean comparisons. The different letters were expressed as significant difference (*p* < 0.05).

The HS-GC-IMS data was processed using a Laboratory Analytical Viewer (LAV, G.A.S., Dortmund, Germany) with three plug-ins and GC × IMS Library Search (NIST database and IMS database). Topographic plots and fingerprints of volatile compounds were established by plug-ins with Reporter and Gallery Plot (G.A.S., Dortmund, Germany). Principal component analysis (PCA) was performed using the Dynamic PCA plug-in (G.A.S., Dortmund, Germany) to evaluate the regularity and difference among tested samples. All the measurements were performed in triplicate.

## 4. Conclusions

In this study, Cuiguan pears from different geographical areas had been observed for significant difference in VOCs; physical and chemical parameters; and sensory. GC-IMS results showed that 50 volatiles compounds were identified, including esters (25), alcohols (8), aldehydes (8), ketones (5), terpenes (2), thiophene (1) and sulfides (1). The PCA results indicated that differences in volatiles among the samples from different regions were evident. In sensory analysis, the ‘ZHs’ obtained the higher average scores of acceptance from consumers than other pears. Result of sensory reflected the difference among the pears from different geographical areas. It emerged that most of the parameters with appropriate statistics allowed distinguishment of the four pears. For example, GC-IMS combined with PCA was an efficient method to distinguish different regions’ pears. Overall, our findings may provide some theoretical basis for the improvement and development of Cuiguan pear variety breeding, planting conditions and quality enhancement in the future.

## Figures and Tables

**Figure 1 molecules-28-01733-f001:**
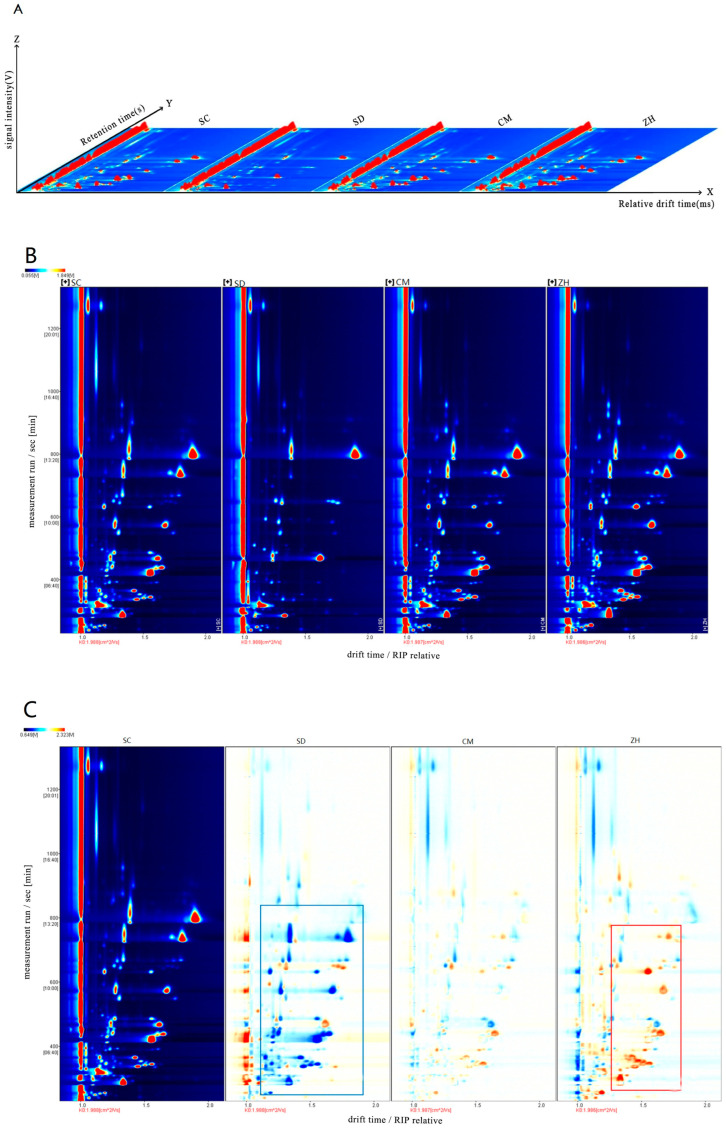
GC-IMS analysis of four different Cui Guan pears. (**A**) 3D-topographic; (**B**) 2D-topographic plots; (**C**) The difference comparison topographic plots.

**Figure 2 molecules-28-01733-f002:**
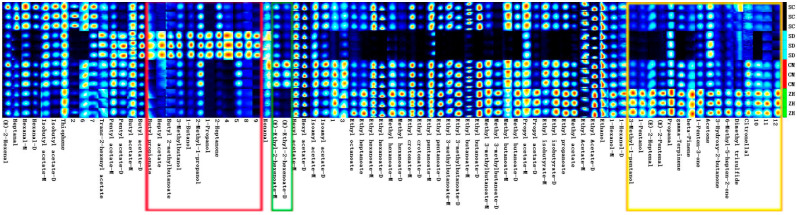
Fingerprints of volatile compounds in Cuiguan pears with different regions.

**Figure 3 molecules-28-01733-f003:**
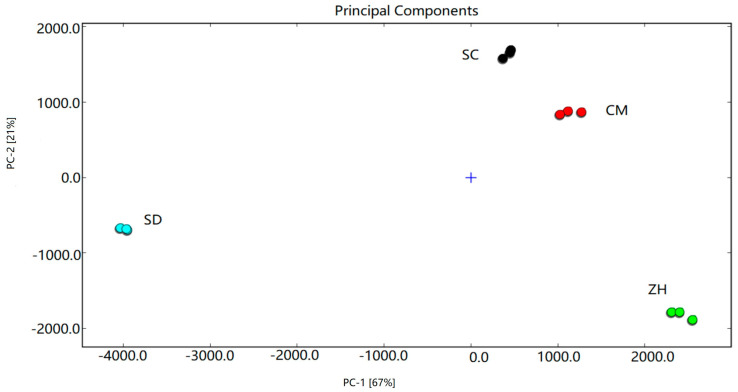
PCA analysis of Cui Guan pear samples with different regions.

**Figure 4 molecules-28-01733-f004:**
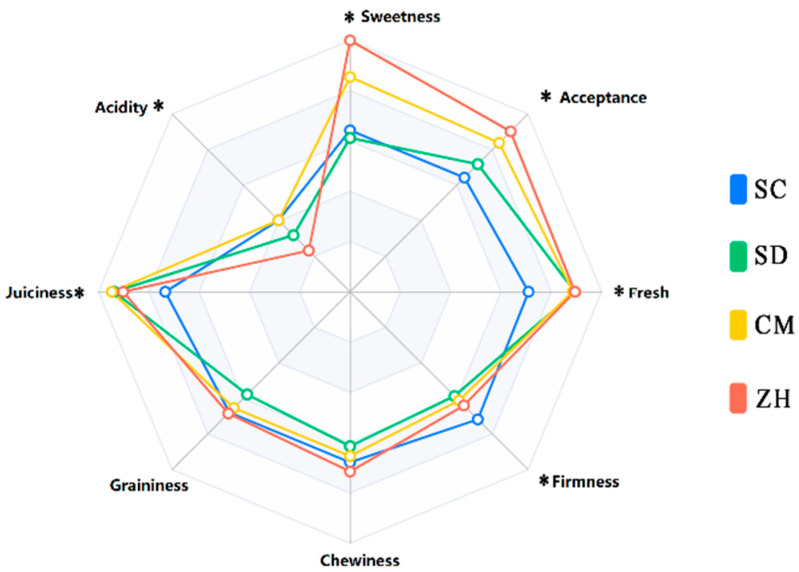
Schematic representation of the average values of the scores of the eight attributes of consumer sensory analysis (‘*’ *p* < 0.05).

**Table 1 molecules-28-01733-t001:** The physicochemical parameters of Cuiguan pear with different regions.

Physicochemical	SC	SD	CM	ZH
Fruit shape	0.92 ± 0.054 a	0.92 ± 0.053 a	0.94 ± 0.072 a	0.97 ± 0.056 a
weight/g	279.72 ± 29.02 b	302.27 ± 35.72 ab	306.17 ± 23.79 ab	317.78 ± 18.74 a
Firmness/N	16.69 ± 3.12 a	12.12 ± 1.10 b	13.65 ± 1.77 b	16.20 ± 0.64 a
Lightness	56.06 ± 2.97 b	59.67 ± 1.98 a	51.50 ± 3.47 d	53.91 ± 1.50 c
Chroma	30.96 ± 1.54 a	29.45 ± 0.93 bc	28.92 ± 1.70 c	30.23 ± 0.82 ab
Hue angle	115.50 ± 3.94 b	121.56 ± 0.83 a	116.58 ± 5.54 b	120.29 ± 1.58 a
Vitamin C mg/100 g	3.60 ± 0.16 c	2.26 ± 0.16 d	6.88 ± 0.23 a	5.80 ± 0.60 b
TSS/°Brix	13.18 ± 0.15 d	13.78 ± 0.15 c	16.08 ± 0.15 a	14.66 ± 0.050 b
TA mg/g	4.54 ± 0.077 c	4.60 ± 0.063 c	4.99 ± 0.22 b	5.31 ± 0.083 a
TP mg/g	2.83 ± 0.0069 a	1.20 ± 0.022 c	2.64 ± 0.0069 b	2.63 ± 0.029 b
TFA mg/g	0.92 ± 0.10 c	0.62 ± 0.0058 d	1.14 ± 0.015 b	1.26 ± 0.10 a

Different letters indicate statistical differences within Cuiguan pears, according to SNK test (*p* < 0.05).

**Table 2 molecules-28-01733-t002:** The mineral element of Cuiguan pear with four different regions (mg/kg).

Mineral Element	SC	SD	CM	ZH
K	893.31 ± 32.21 c	1238.73 ± 41.36 b	1301.43 ± 13.85 b	1886.73 ± 110.01 a
Mg	65.71 ± 4.07 c	92.08 ± 6.23 ab	80.35 ± 0.074 b	102.84 ± 6.08 a
Ca	49.14 ± 1.73 a	32.79 ± 0.44 c	36.39 ± 1.01 b	26.12 ± 0.013 d
Na	1.97 ± 0.12 c	1.71 ± 0.11 c	3.79 ± 0.043 a	3.34 ± 0.20 b
B	1.96 ± 0.062 b	1.45 ± 0.055 c	2.54 ± 0.036 a	2.60 ± 0.12 a
Fe	1.11 ± 0.056 b	1.19 ± 0.066 b	0.87 ± 0.073 c	1.52 ± 0.068 a
Mn	0.452 ± 0.019 c	0.635 ± 0.018 b	1.019 ± 0.0087 a	1.070 ± 0.058 a
Cu	0.3782 ± 0.02974 b	0.3947 ± 0.00299 b	0.8463 ± 0.03826 a	0.3605 ± 0.00055 b
Zn	0.2972 ± 0.02233 d	0.5431 ± 0.02165 c	0.8194 ± 0.02577 b	1.2009 ± 0.05714 a
Ti	0.0852 ± 0.00324 a	0.0708 ± 0.00529 b	0.0643 ± 0.00507 b	0.0616 ± 0.00623 b
Co	0.0153 ± 0.00054 b	0.0098 ± 0.00044 c	0.0193 ± 0.00024 a	0.0144 ± 0.00084 b
Ni	0.0143 ± 0.00066 c	0.0254 ± 0.00049 c	0.0567 ± 0.00285 b	0.1362 ± 0.10000 a
Mo	0.0090 ± 0.00052 b	0.0051 ± 0.00009 c	0.0020 ± 0.00020 d	0.0111 ± 0.00048 a
Cr	0.0068 ± 0.00052 b	0.0031 ± 0.00038 c	0.0080 ± 0.00087 b	0.0130 ± 0.00111 a
Sn	0.0058 ± 0.00010 a	0.0036 ± 0.00043 b	0.0026 ± 0.00034 b	0.0031 ± 0.00040 b
V	0.0009 ± 0.00003 a	0.0004 ± 0.00006 c	0.0004 ± 0.00004 c	0.0008 ± 0.00005 b
Se	0.0001 ± 0.00002 c	0.0005 ± 0.00000 a	0.0002 ± 0.00001 c	0.0004 ± 0.00004 b

Different letters indicate statistical differences within Cuiguan pears, according to SNK test (*p* < 0.05).

**Table 3 molecules-28-01733-t003:** Volatile compounds tested from Cuiguan pear of different regions using GC–IMS.

	NO.	Compounds	RI	RT/s	DT/ms	Peak Intensity			
						SC	SD	CM	ZH
Esters	1	Ethyl octanoate	1450.3	1141.106	1.49164	530.05 ± 67.21	209.87 ± 50.25	529.11 ± 93.17	841.57 ± 183.67
	2	Ethyl heptanoate	1341.3	900.643	1.41748	885.91 ± 79.55	106.13 ± 8.41	636.28 ± 12.62	1696.07 ± 110.96
	3	Trans-2-hexenyl acetate	1344.4	906.76	1.86815	158.81 ± 39.08	290.83 ± 28.48	183.65 ± 33.03	228.06 ± 24.50
	4	(E)-Ethyl-2-hexenoate-M	1327.9	874.952	1.33037	993.49 ± 202.49	994.01 ± 30.7	1892.69 ± 127.96	520.94 ± 52.83
	5	(E)-Ethyl-2-hexenoate-D	1327.9	874.851	1.81774	229.64 ± 49.54	183.67 ± 14.97	592.97 ± 79.26	104.91 ± 15.87
	6	Hexyl acetate-M	1283.3	795.125	1.38766	12,848.97 ± 205.3	12,359.64 ± 39.65	12,162.45 ± 29.61	11,593.91 ± 98.63
	7	Hexyl acetate-D	1282.7	794.141	1.89942	29,366.12 ± 1646.5	25,164.38 ± 137.24	29,975.07 ± 277.23	24,337.57 ± 1186.5
	8	Ethyl hexanoate-M	1246	736.069	1.34099	10,080 ± 257.58	411.49 ± 76.35	9598.05 ± 9.21	9709.26 ± 58.46
	9	Ethyl hexanoate-D	1245.4	735.084	1.79773	15,187.34 ± 1295.06	286.79 ± 106.54	14,115.22 ± 446.82	18,241.16 ± 1156.04
	10	Methyl hexanoate-M	1197.1	665.126	1.28589	672.84 ± 9.29	187.57 ± 7.22	1126.9 ± 74.96	1473.09 ± 107.09
	11	Methyl hexanoate-D	1197.4	665.5	1.68002	215.33 ± 20.56	54.78 ± 7.14	348.18 ± 64.82	687.81 ± 116.74
	12	Pentyl acetate-M	1185.7	644.583	1.31327	934.5 ± 71.98	2140.78 ± 113.43	1975.34 ± 37.74	1411.42 ± 49.76
	13	Pentyl acetate-D	1185.7	644.583	1.76502	147.69 ± 14.78	931.53 ± 155.23	654.3 ± 15.39	421.77 ± 18.81
	14	Ethyl crotonate-M	1179.5	631.329	1.18702	2893.51 ± 75.62	287.23 ± 8.60	3390.91 ± 84.9	4262.17 ± 17.19
	15	Ethyl crotonate-D	1179.8	631.992	1.55512	1462.16 ± 45.11	114.85 ± 30.63	2432.14 ± 163.52	8292.16 ± 296.9
	16	Ethyl pentanoate-M	1150.3	573.013	1.27372	5775.85 ± 128.67	1672.96 ± 51.12	5607.58 ± 13.59	5846.96 ± 36.27
	17	Ethyl pentanoate-D	1150.6	573.676	1.68289	7233.27 ± 670.83	317.89 ± 53.46	7265.86 ± 86.56	13,328.21 ± 411.47
	18	Isoamyl acetate-M	1137.1	548.494	1.3011	2003.05 ± 34.58	1412.63 ± 130.10	2205.92 ± 46.55	1702.05 ± 72.54
	19	Isoamyl acetate-D	1137.5	549.157	1.74373	515.54 ± 14.5	226.24 ± 36.77	662.05 ± 47.53	413.86 ± 41.00
	20	Butyl acetate-M	1087.7	467.162	1.23797	3188.14 ± 51.35	4242.24 ± 15.16	2962.95 ± 35.86	2680.29 ± 16.65
	21	Butyl acetate-D	1088.1	467.602	1.61833	8129.94 ± 478.78	13,227.47 ± 337.77	5015.73 ± 90.78	6066.65 ± 214.5
	22	Ethyl 3-methylbutanoate-M	1067.4	439.87	1.25011	2533.08 ± 33.8	86.69 ± 4.53	2631.37 ± 21.66	2163.27 ± 45.09
	23	Ethyl 3-methylbutanoate-D	1067.1	439.43	1.6534	5045.97 ± 170.28	103.52 ± 21.38	9213.3 ± 105.43	11,189.66 ± 210.06
	24	Ethyl butanoate-M	1051.9	420.061	1.2056	2657.61 ± 26.81	367.92 ± 102.66	2358.66 ± 46.04	1658.96 ± 24.15
	25	Ethyl butanoate-D	1051.9	420.061	1.56303	22,033.53 ± 399.31	120.89 ± 48.72	22,866.79 ± 85.21	25,022.45 ± 119.15
	26	Isobutyl acetate-M	1027.7	391.009	1.22718	777.78 ± 13.57	691.76 ± 36.90	620.71 ± 2.17	444.61 ± 18.42
	27	Isobutyl acetate-D	1027.7	391.009	1.61024	311.8 ± 17.24	162.12 ± 16.47	188.59 ± 1.33	77.09 ± 5.19
	28	Methyl 3-methylbutanoate-M	1023.9	386.607	1.19481	367.8 ± 3.68	82.24 ± 0.27	892.71 ± 42.34	935.42 ± 27.83
	29	Methyl 3-methylbutanoate-D	1024.8	387.673	1.52871	124.73 ± 4.33	33.85 ± 5.63	500.3 ± 68.14	906.25 ± 89.67
	30	Methyl butanoate-M	1004.2	364.701	1.15204	1397.28 ± 30.04	98.31 ± 14.21	1360.65 ± 15.02	953.67 ± 20.93
	31	Methyl butanoate-D	1004.6	365.054	1.4365	3450.83 ± 361.74	38.43 ± 7.63	4987.69 ± 246.49	6220.27 ± 164.31
	32	Propyl acetate-M	992.9	353.408	1.16539	650.79 ± 14.89	501.4 ± 49.38	511.35 ± 30.03	614.42 ± 4.10
	33	Propyl acetate-D	992.9	353.408	1.47523	989.16 ± 26.16	221.56 ± 33.98	752.8 ± 88.51	2847.89 ± 84.56
	34	Ethyl isobutyrate-M	983.0	345.644	1.20145	1057.51 ± 31.99	110.27 ± 3.19	1225.46 ± 30.67	670.06 ± 11.49
	35	Ethyl isobutyrate-D	983.0	345.644	1.56204	2300.76 ± 70.06	52.08 ± 16.77	4306.63 ± 112.19	5189.3 ± 105.41
	36	Ethyl propanoate	971.5	336.821	1.44852	4449.82 ± 174.6	144.65 ± 11.12	5656.43 ± 91.58	7885.22 ± 35.17
	37	Ethyl Acetate-M	896.2	284.236	1.09862	2733.14 ± 53.91	2733.08 ± 10.1	2678.67 ± 33.58	1505.43 ± 47.87
	38	Ethyl Acetate-D	897.3	284.942	1.33366	18,153.65 ± 239.6	7412.66 ± 234.08	19,089.9 ± 503.3	28,008.74 ± 450.67
	39	Methyl acetate	848.5	255.297	1.19077	1437.36 ± 43.13	416.82 ± 20.05	2544.66 ± 207.13	5668.96 ± 252.90
	40	Butyl propionate	1155.7	583.347	1.71798	68.57 ± 12.91	146.23 ± 5.11	35.07 ± 2.26	82.96 ± 6.49
	41	Heptyl acetate	1382.9	985.719	1.46745	55.47 ± 18.76	202.81 ± 9.34	103.08 ± 31.14	71.55 ± 4.95
	42	Butyl 2-methylbutanoate	1239.6	726.243	1.3887	392.16 ± 27.79	716.7 ± 142.18	396.25 ± 21.5	440.67 ± 5.82
Alcohols	43	1-Hexanol-M	1369.2	956.83	1.33015	1225.45 ± 207.89	896.59 ± 46.8	1469.13 ± 27.85	1914.18 ± 153.08
	44	1-Hexanol-D	1369.2	956.919	1.64092	107.36 ± 40.99	84.6 ± 24.78	127.94 ± 20.49	149.8 ± 25.44
	45	3-Methyl-1-pentanol	1353.0	923.887	1.33037	256.58 ± 12.57	59.39 ± 16.41	330.81 ± 15.42	1168.11 ± 41.8
	46	3-Methylbutanol	1220.6	698.26	1.2433	109.92 ± 13.69	240.94 ± 13.9	151.95 ± 8.62	105.55 ± 1.78
	47	1-Butanol	1160.2	592.231	1.18093	232.33 ± 13.13	791.62 ± 26.2	217.28 ± 1.3	235.48 ± 6.49
	48	2-Methyl-1-propanol	1107.2	496.654	1.17188	485.92 ± 3.72	674.28 ± 21.46	606.16 ± 7.00	382.6 ± 8.9
	49	Ethanol	944.2	316.7	1.13067	16,200.12 ± 63.48	14,982.91 ± 225.95	15,946.15 ± 36.82	16,400.76 ± 84.63
	50	1-Propanol	1053.1	421.521	1.11064	19.28 ± 6.71	479.78 ± 30.78	16.25 ± 4.72	18.91 ± 1.73
	51	1-Pentanol	1270.1	773.604	1.25575	159.55 ± 20.58	57.26 ± 11.54	164.77 ± 16.77	503.61 ± 26.72
Aldehydes	52	Nonanal	1402.2	1027.929	1.48413	223.63 ± 47.89	285.15 ± 30.25	314.21 ± 62.61	242.65 ± 9.76
	53	(E)-2-Heptenal	1316.9	854.181	1.25931	541.59 ± 52.58	62.66 ± 18.71	746.49 ± 39.38	1264.08 ± 61.98
	54	(E)-2-Hexenal	1231.6	714.415	1.18429	516.44 ± 132.74	330.78 ± 8.36	563.42 ± 64.86	481.55 ± 39.67
	55	Heptanal	1200.0	669.138	1.33432	2635.14 ± 353.37	377.35 ± 46.41	1262.81 ± 59.65	1478.24 ± 218.15
	56	(E)-2-Pentenal	1125.2	527.288	1.12313	115 ± 16.88	45.51 ± 17.70	164.6 ± 23.82	411.36 ± 26.97
	57	Hexanal-M	1100.4	485.649	1.27034	1965.6 ± 218.09	620.77 ± 22.61	1510.94 ± 32.78	990.05 ± 56.57
	58	Hexanal-D	1100.4	485.649	1.56573	2251.42 ± 719.96	410.7 ± 37.30	1163.73 ± 74.16	1270.04 ± 125.48
	59	Propanal	819.4	239.062	1.0679	1125.01 ± 22.17	1399.97 ± 26.79	1272.33 ± 2.81	2222.32 ± 71.13
	60	Citronellal	1501.0	1273.853	1.34105	286.78 ± 79.92	202.4 ± 93.15	389.98 ± 51.55	477.62 ± 20.32
Ketones	61	2-Heptanone	1188.5	650.547	1.26307	888.51 ± 37.58	1364.16 ± 14.66	234.06 ± 13.38	303.47 ± 25.93
	62	1-Penten-3-one	1030.0	393.65	1.09365	290.96 ± 16.36	164.98 ± 3.11	242.63 ± 10.82	427.09 ± 18.2
	63	Acetone	837.3	248.944	1.11331	822.76 ± 85.5	902.13 ± 92.97	653.2 ± 7.51	916.91 ± 27.5
	64	3-Hydroxy-2-butanone	1298.1	820.068	1.06583	205.78 ± 17.96	198.77 ± 12.32	261.04 ± 7.37	531.42 ± 46.03
	65	6-Methyl-5-hepten-2-one	1345.9	909.795	1.18252	121.29 ± 15.45	107.26 ± 13.67	132.37 ± 13.55	257.2 ± 38.8
Thiophene	66	Thiophene	1030.3	393.993	1.0452	1279.36 ± 78.31	1175.64 ± 64.91	1061.56 ± 27.86	1301.33 ± 28.93
Terpenes	67	beta-Pinene	1103.6	490.867	1.21641	211.47 ± 31.93	67.03 ± 1.00	318.06 ± 11.35	556.45 ± 22.74
	68	gamma-Terpinene	1248.6	740.009	1.71635	1934.41 ± 73.68	74.16 ± 23.13	3222.67 ± 8.67	4346.54 ± 157.79
Sulfides	69	Dimethyl trisulfide	1345.9	909.795	1.18252	180.25 ± 3.59	237.82 ± 35.15	163.47 ± 17.14	436.84 ± 28.80

‘M’ and ‘D’ mean the monomer and dimer of the volatile; RI means the retention index of the volatiles in a wax capillary column; RT means the retention time of the volatiles in GC-IMS; DT means the drift time of the volatiles in GC-IMS.

## Data Availability

Not applicable.

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
