# Peer review of "Comparative Analysis of the Quality in Ripe Fruits of Cuiguan Pear from Different Regions"

_molecules, 2023, doi:10.3390/molecules28041733_

Round 1

Reviewer 1 Report

The authors reported the comparative analysis results of Cuiguan pears from four different regions mainly by GC-IMS and sensory analysis. However, these methods are the same as those in reference 28 (Food Chem. X 2022, 15, 100423, doi:10.1016/j.fochx.2022.100423.), which was reported in 2022 by the same authors of this manuscript. Another major drawback of this manuscript is the lack of bioactivities tested of the high content volatile compounds, although the difference between four origins were found. Line 261, how to determine the numbers of people to assess? How to select people, because people in different places may have different sensory feelings? And also some grammatical errors should be corrected to promote the conceptual flow. In general, the result of this manuscript was not important in the field. This manuscript does not meet the required quality standards of Molecules.

Author Response

Point 1: The authors reported the comparative analysis results of Cuiguan pears from four different regions mainly by GC-IMS and sensory analysis. However, these methods are the same as those in reference 28 (Food Chem. X 2022, 15, 100423, doi:10.1016/j.fochx.2022.100423.), which was reported in 2022 by the same authors of this manuscript. Another major drawback of this manuscript is the lack of bioactivities tested of the high content volatile compounds, although the difference between four origins were found. Line 261, how to determine the numbers of people to assess? How to select people, because people in different places may have different sensory feelings? And also some grammatical errors should be corrected to promote the conceptual flow. In general, the result of this manuscript was not important in the field. This manuscript does not meet the required quality standards of Molecules.

Response 1:

Thank you for raising this valuable question.

The purpose of this paper is to detect and analyze the physical and chemical indexes and volatile substances of four kinds of Cuiguan pears from different places of origin, find out the differences in the quality and flavor of different cuiguan pears, and provide certain data reference for variety selection, quality improvement and brand building. In, recent years, HS-GC-IMS has been applied in food flavor analysis due to its advantages. We studied the flavor of Cuiguan pears from different geographical sources by HS-GC-IMS in this study.  

The high content of bioactive substances has not been tested and analyzed, with the in-depth study of the nutritional quality of Cuiguan pear, we may carry out relevant research in the future.

For the selection of sensory personnel, we mainly consider the general and objective sensory reflection of the four kinds of Cuiguan pears, without defining the characteristics of the personnel. The sensory personnel are composed of 70 people, aged between 20 and 60 years old, including researchers and inspectors from Institute of Forestry and Fruit, Institute of Agricultural Product Quality Standards and Testing Technology, Shanghai Academy of Agricultural Sciences, as well as undergraduate interns and graduate students. All sensory personnel may come from different regions and have not been trained in sensory evaluation before.

Reviewer 2 Report

This manuscript reports volatile organic compounds, sensory and physicochemical parameters of Cuiguan pear from four different regions of China. In general, the results may provide some reference for the aroma of pears could be affected by origin areas.

Major comments

Plant materials.  “All pears were collected at maturity stage.” How did the authors define the “maturity stage”? Pears at different maturity stages obviously have different volatile compounds. The authors should provide the pictures of Cuiguan pear at different maturity stages, and mark the sampling stage for this experiment.

Volatile organic compounds analysis. The pear flesh samples (2.0g) were for HS-GC-IMS analysis of VOCs of Cuiguan pear. Since different flesh samples from different locations in a pear fruit may affect the VOC results, the authors should provide a detailed description of sampling method and indicate the sample part in the fruit. In addition, “All measurements were carried out on three technical replicates.” Why did not use biological replicates?

Minor comments.

Line 2, change “comparative” to “Comparative”.

Line 14, change “Zhuanghang (ZH))” to “Zhuanghang (ZH)”.

Line 31-32, “Pear is one of the most economically important temperate fruit tree species, with high economic value[1]”. “economically important” is the same meaning with “with high economic value”.

Line 35-36, “As one of the origin center of pear species, pears are the third-most important fruit in China[3].” China as one of the origin center of pear species, not the pears. This sentence need to be reworded.

Line 36, change “variety” to “cultivar”

Line 40, generally, for kiwifruits, fruit aroma is not a key factor to evaluate kiwifruits and affect consumers' purchase intention. Thus, change the reference [6] to a recent review “Floral Scents and fruit aromas: functions, compositions, biosynthesis, and regulation, Frontiers in Plant Science, 2022, 13:860157”.

Table 1, what are “L”, “C”, and “h” in the physicochemical parameters? They should be unified to use their full names.  “Vitamin C,” delete the comma.

Line 201, “meterials” should be “materials”.

Author Response

Point 1: Plant materials.  “All pears were collected at maturity stage.” How did the authors define the “maturity stage”? Pears at different maturity stages obviously have different volatile compounds. The authors should provide the pictures of Cuiguan pear at different maturity stages, and mark the sampling stage for this experiment.

Response 1: We thank the reviewers for raising this question. All pears were collected at the same time and are mature, freshly picked and marketed products. For each variety, all pears collected in this study are of similar size, without visible external damage.

Point 2: Volatile organic compounds analysis. The pear flesh samples (2.0g) were for HS-GC-IMS analysis of VOCs of Cuiguan pear. Since different flesh samples from different locations in a pear fruit may affect the VOC results, the authors should provide a detailed description of sampling method and indicate the sample part in the fruit. In addition, “All measurements were carried out on three technical replicates.” Why did not use biological replicates?

Response 2: Thank you for raising this valuable question. The pear flesh samples of each biological duplication are obtained from 0.5 cm under the skin of the pear at the upper, middle and lower parts of the pear, and then tested after being crushed evenly. We use biological repetition. Every pear is biological repetition. HS-GC-IMS analysis of volatile organic compounds was conducted in three technical and biological replicates.

Point 3: Line 2, change “comparative” to “Comparative”.

Line 14, change “Zhuanghang (ZH))” to “Zhuanghang (ZH)”.

Line 31-32, “Pear is one of the most economically important temperate fruit tree species, with high economic value[1]”. “economically important” is the same meaning with “with high economic value”.

Line 35-36, “As one of the origin center of pear species, pears are the third-most important fruit in China[3].” China as one of the origin center of pear species, not the pears. This sentence need to be reworded.

Line 36, change “variety” to “cultivar”

Line 40, generally, for kiwifruits, fruit aroma is not a key factor to evaluate kiwifruits and affect consumers' purchase intention. Thus, change the reference [6] to a recent review “Floral Scents and fruit aromas: functions, compositions, biosynthesis, and regulation, Frontiers in Plant Science, 2022, 13:860157”.

Table 1, what are “L”, “C”, and “h” in the physicochemical parameters? They should be unified to use their full names.  “Vitamin C,” delete the comma.

Line 201, “meterials” should be “materials”.

Response 3: Thank the reviewer for raising this question. For the above modification suggestions, the manuscript was revised.

Round 2

Reviewer 1 Report

 Accept in present form.

Reviewer 2 Report

All my questions and concerns have been addressed.